# The memory of airway epithelium damage in smokers and COPD patients

François M Carlier[1,2,3], Bruno Detry[1], Marylène Lecocq[1], Amandine M Collin[1], Thomas Planté-Bordeneuve[1], Ludovic Gérard[1], Stijn E Verleden[4], Monique Delos[5], Benoît Rondelet[3,6], Wim Janssens[4], Jérôme Ambroise[7], Bart M Vanaudenaerde[4], Sophie Gohy[1,8,9], Charles Pilette[1,8]

Chronic obstructive pulmonary disease (COPD), a devastating and irreversible lung disease, causes structural and functional defects in the bronchial epithelium, the (ir)reversibility of which remains unexplored in vitro. This study aimed to investigate the persistence of COPD-related epithelial defects in long-term airway epithelial cultures derived from non-smokers, smokers, and COPD patients. Barrier function, polarity, cell commitment, epithelial-to-mesenchymal transition, and inflammation were evaluated and compared with native epithelium characteristics. The role of inflammation was explored using cytokines. We show that barrier dysfunction, compromised polarity, and lineage abnormalities observed in smokers and COPD persisted for up to 10 wk. Goblet cell hyperplasia was associated with recent cigarette smoke exposure. Conversely, increased IL-8/CXCL-8 release and abnormal epithelial-to-mesenchymal transition diminished over time. These ex vivo observations matched surgical samples' abnormalities. Cytokine treatment induced COPD-like changes in control cultures and reactivated epithelial-to-mesenchymal transition in COPD cells. In conclusion, these findings suggest that the airway epithelium of smokers and COPD patients retains a multidimensional memory of its original state and previous cigarette smoke-induced injuries, maintaining these abnormalities for extended periods.

## Introduction

Chronic obstructive pulmonary disease (COPD) constitutes the third cause of death worldwide (World Health Organization, 2017) and is mainly due to noxious airborne stimuli, most importantly cigarette smoke (Salvi & Barnes, 2009). COPD is characterized by the irreversible narrowing and disappearance of small conducting airways (McDonough et al, 2011) and destruction of alveolar walls (Vogelmeier et al, 2017). These changes lead to airway obstruction (bronchitis) and emphysema, with their relative contribution varying among individuals diagnosed with COPD. Current therapies provide limited clinical and functional benefits and fail at targeting the underlying pathways leading to structural remodeling of the lungs (Celli et al, 2021).

The airway epithelium (AE), as the first-line barrier against inhaled particles, is constantly exposed to airborne pollutants and fulfils multiple functions to maintain pulmonary homeostasis. It ensures adequate barrier function, cell differentiation, and polarization, maintaining a tight control on inflammatory mechanisms. In COPD, the AE fails at these duties, exhibiting altered physical barrier function (Aghapour et al, 2018; Carlier et al, 2021), underpinned by decreased expression of apical junction complexes (AJCs) proteins (Gohy et al, 2015), abnormal differentiation and function of ciliated and club cells (Pilette et al, 2001; Gamez et al, 2015; Yaghi & Dolovich, 2016; Gohy et al, 2019), hyperplastic goblet cells (Saetta et al, 2000), and aberrant epithelial-to-mesenchymal transition (EMT) that is triggered by cigarette smoke (Milara et al, 2013) and further enhanced in COPD (Mahmood et al, 2015). In addition, the polymeric immunoglobulin receptor (pIgR)/secretory component (SC) system, which ensures the transcytosis and release of polymeric immunoglobulins into mucosal secretions, witnessing epithelial polarization, is also defective in COPD (Gohy et al, 2014; Carlier et al, 2020), with decreased epithelial pIgR expression (Gohy et al, 2014) and local S-IgA deficiency in small airways (Polosukhin et al, 2011, 2017). Finally, epithelial inflammation control is dysregulated in COPD (Barnes, 2016), with increased intraepithelial neutrophils (Eapen et al, 2017) and sputum TNF-α, IL-8/CXCL-8, and neutrophils (Keatings et al, 1996; Rutgers et al, 2000).

The air/liquid interface (ALI) culture model of primary human bronchial epithelial cells (HBEC) allows the reconstitution of the AE

[1]Pole of Pneumology, ENT, and Dermatology, Institute of Experimental and Clinical Research, Université Catholique de Louvain, Brussels, Belgium    [2]Department of Pneumology, CHU Mont-Godinne UCL Namur, Yvoir, Belgium    [3]Lung Transplant Centre, CHU Mont-Godinne UCL Namur, Yvoir, Belgium    [4]Department of Chronic Diseases, Metabolism and Ageing, Katholieke Universiteit Leuven, Leuven, Belgium    [5]Department of Pathology, CHU Mont-Godinne UCL Namur, Yvoir, Belgium    [6]Deparment of Cardiovascular and Thoracic Surgery, CHU Mont-Godinne UCL Namur, Yvoir, Belgium    [7]Centre de Technologies Moléculaires Appliquées, Institute of Experimental and Clinical Research, Université Catholique de Louvain, Brussels, Belgium    [8]Department of Pneumology, Cliniques Universitaires St-Luc, Brussels, Belgium    [9]Cystic Fibrosis Reference Center, Cliniques Universitaires St-Luc, Brussels, Belgium

Correspondence: carlierfrancois@gmail.com

in vitro and recapitulates several of the alterations observed in situ in COPD. Cigarette smoke-exposed COPD HBEC show decreased E-cadherin and ZO-1 expression and reduced transepithelial electric resistance (TEER) when compared with smokers without COPD (Milara et al, 2013; Heijink et al, 2014) and after 4 wk culture, the COPD ALI-AE displays altered lineage differentiation, decreased pIgR expression, and EMT (Gohy et al, 2014, 2015, 2019), along with increased cytokine release (Schneider et al, 2010). Although these data suggest the persistence of AE abnormalities, it remains unclear whether (and to what extent) these structural changes persist on the long-term, matching the irreversible nature of COPD. COPD patients who quit smoking for more than 3.5 yr display less goblet cell hyperplasia (Lapperre et al, 2007), and smoking cessation may improve lung function and survival (Willemse et al, 2004; Bai et al, 2017). Conversely, smoking cessation does not influence COPD-related epidermal growth factor receptor activation (Lapperre et al, 2007) or protease activity (Louhelainen et al, 2009). In COPD, the inflammatory pattern shared with "healthy" smokers is amplified and persists even after smoking cessation, although with conflicting data (Willemse et al, 2004; Lapperre et al, 2006; Gamble et al, 2007).

Whereas small airway changes are importantly involved in COPD pathophysiology, the recent *Lancet Commission* advocates for a broader definition including symptoms related to large airways (e.g., bronchitis) and for active research in the field (Stolz et al, 2022). The present study aims to elucidate whether epithelial changes in large airways are persistently imprinted, questioning the disease memory (Ordovas-Montanes et al, 2020) retained in the COPD epithelium. To address this question, we reconstituted HBEC-derived ALI-AE from non-smokers, non-COPD smokers, and COPD patients and cultured it in vitro for up to 10 wk. The spontaneous evolution of abnormalities was assayed for epithelial readouts including barrier function, cell differentiation, EMT, pIgR/SC-related polarity, and production of inflammatory cytokines. In addition, we assessed whether exogenous inflammation could trigger COPD-related changes.

# Results

Readouts were assessed (for each sample, according to availability) at different timepoints of redifferentiation, referred to as early (1 wk ALI), short-term (2–3 wk), mid-term (4–7 wk), and long-term (8–10 wk) cultures.

## Barrier and junctional properties in the COPD AE

The AE physical barrier was assessed by measuring TEER of ALI-AE up to 10 wk. According to smoking status and lung function, the study population was divided into four groups: non-smoker controls (NS; n = 5), smoker controls (Smo, n = 7), mild/moderate COPD (COPD1-2; n = 7), and severe/very severe COPD (COPD3-4; n = 7). The characteristics of the study population are summarized in Table 1.

The COPD AE displayed decreased TEER as compared with NS and, to a lesser extent, with Smo, which also showed decreased TEER compared with NS (Fig 1A). This defect appeared in early

cultures, persisting in long-term cultures (Fig 1B). In addition, TEER was inversely correlated with the disease severity witnessed by the forced expiratory volume in 1 s (FEV1). This correlation was significant from early up to long-term cultures (Fig 1C).

The molecular substratum of this long-lasting barrier disruption was investigated by assessing mRNA abundance and protein expression of major components of AJCs, namely claudin-1 (*CLDN1*), E-cadherin (*CDH1*), occludin (*OCLN*), and ZO-1/tight junction protein 1 (*TJP1*). Although no difference was observed regarding mRNA abundance (Fig S1), protein expression of E-cadherin was reduced in COPD versus NS in early and short-term ALI (Fig 2A), whereas that of occludin was decreased from early up to long-term cultures in Smo and COPD (Fig 2B) and inversely correlated with FEV1 up to mid-term (Fig 2C). Fig 2D shows representative blots for E-cadherin and occludin in NS and COPD3-4 AE.

These data globally depict epithelial barrier dysfunction that is engaged in Smo, further worsens in COPD, and persists upon prolonged culture.

## Lineage differentiation of the COPD AE

The persistence of differentiation alterations in COPD was explored by assessing specific markers and transcription factors of early differentiation towards intermediate cells, and of goblet and ciliated cells.

### Basal cells early differentiation
mRNA abundance of MYB Proto-Oncogene (*MYB*), a marker of early differentiation of basal cells (Pan et al, 2014), was decreased up to long-term in Smo and COPD-derived AE (Fig S2A), correlating with FEV1 at some time-periods (Fig S2B).

### Differentiation towards ciliated cells
The differentiation towards ciliated cells was assessed by measuring the mRNA abundance of Forkhead Box J1 (*FOXJ1*), a transcription factor involved in ciliated cells' commitment, and of the dynein axonemal intermediate chain-1 (*DNAI1*), a marker of ciliated cells' terminal differentiation. We also counted $\beta$-tubulin IV$^+$ cells in ALI-AE and in native tissues. Smo- and COPD-derived cultures displayed persisting decreases of both gene transcripts, which correlated with FEV1 at some time periods (Figs 3A and B and S3A and B). $\beta$-tubulin IV$^+$ (ciliated) cell numbers were decreased up to long-term ALI-cultures from Smo and COPD (Fig 3C and D), in line with reduced $\beta$-tubulin IV$^+$ surface in the native COPD AE (Fig 3E).

### Differentiation towards goblet cells
The mRNA abundance of SAM Pointed Domain Containing ETS Transcription Factor (*SPDEF*) and Forkhead Box A3 (*FOXA3*) was assessed as transcription factors inducing goblet cell differentiation (Park et al, 2007; Chen et al, 2014; Rajavelu et al, 2015), along with the protein expression of MUC5AC, a terminal product of goblet cell differentiation. Compared with NS, *SPDEF* expression was persistently increased in Smo and in COPD, although to a lesser extent (Fig 4A). Similarly, MUC5AC$^+$ goblet cell numbers were increased in Smo ALI-cultures versus NS (Fig 4B), matching in situ findings, where MUC5AC$^+$ surface in Smo-AE was considerably larger than in

**Table 1.   Patient cohort for ALI cultures.**

| | Non-smoker controls (n = 5) | Smoker controls (n = 7) | COPD1-2 (n = 7) | COPD3-4 (n = 7) | |
|---|---|---|---|---|---|
| N (male/female) | 5 (2/3) | 7 (6/1) | 7 (4/3) | 7 (2/5) | ns |
| Age (yr) | 67.8 ± 13.8 | 66.1 ± 11.1 | 66.4 ± 12.3 | 58.8 ± 5.7 | ns |
| Smoking history (never/former/current n) | 5/0/0 | 0/5/2 | 0/1/6 | 0/7/0 | $P = 0.001$ |
| Pack-years | NA | 28.6 ± 14.4 | 43.5 ± 15.6 | 34.4 ± 10.8 | ns |
| If applicable, duration since smoking cessation (months) | NA | 73.1 ± 98.4 | 14.5 ± 23.9 | 75.4 ± 57.0 | ns |
| FEV1 (% of PV) | 100.0 ± 12.7 | 94.5 ± 18.9 | 77.3 ± 10.4[a,b] | 21.0 ± 2.3[a,b,c] | $P < 0.0001$ |
| FEV1/VC ratio (%) | 81.8 ± 4.7 | 83.4 ± 6.5 | 54.6 ± 6.9[a,b] | 31.6 ± 5.9[a,b,c] | $P < 0.0001$ |
| DLCO (% of PV) | 97.4 ± 13.0 | 90.2 ± 13.7 | 59.2 ± 13.1[a] | 36.6 ± 8.0[a,b,c] | $P < 0.0001$ |
| BMI (kg.m$^{-2}$) | 28.9 ± 3.5 | 26.6 ± 5.5 | 23.7 ± 5.1 | 22.5 ± 3.7 | ns |
| Inhaled corticosteroids (n/total N) | 0/5 | 1/7 | 0/7 | 7/7 | $P < 0.0001$ |
| Surgical indication | | | | | |
| Neoplasia | 4/5 | 8/8 | 5/6 | 0/6 | |
| SCC | 0 | 2 | 1 | 0 | |
| AC | 2 | 4 | 3 | 0 | |
| Carcinoid tumor | 2 | 1 | 0 | 0 | $P < 0.0001$[d] |
| Pulmonary metastasis of other cancers | 0 | 1 | 1 | 0 | |
| Lung transplant | 0 | 0 | 0 | 6 | |
| Other | 0 | 0 | 1 | 0 | |
| Declined lung donor | 1 | 0 | 0 | 0 | |

Data are presented as mean ± SD, unless otherwise stated. Demographic data, lung function tests, smoking history, and inhaled corticotherapy are stated for the patient groups, classified according to smoking history and the presence and severity of airflow limitation. AC, adenocarcinoma; ALI, air/liquid interface; BMI, body mass index; COPD, chronic obstructive pulmonary disease; DLCO, diffusing capacity of the lung for CO; FEV1, forced expiratory volume in 1 s; PV, predicted values; SCC, squamous cell cancer; SD, standard deviation; VC, vital capacity; ns, not significant.
[a]$P < 0.05$ compared to non-smoker controls.
[b]$P < 0.05$ compared to smoker controls.
[c]$P < 0.05$ compared to COPD stage 1–2 patients.
[d]Comparison between "Neoplasia," "Lung transplant," and "Other."

NS (Fig 4C). As for *SPDEF*, *FOXA3* was up-regulated in Smo compared with NS (Fig S3C).

Both *SPDEF* and *FOXA3* expression were decreased at all time-points in smokers who quit smoking for more than 4 yr, versus active smokers and ex-smokers with a smoking cessation of less than 4 yr (Fig 4D). These results corroborated in situ findings, with reduced MUC5AC⁺ surface in long-quitters' AE (Fig 4C, right graph).

These data show that goblet cell hyperplasia relates to active/recent smoking and demonstrate its persistence over time in vitro.

### EMT of the COPD AE

The expression of EMT-related protein markers (vimentin, fibronectin) was assessed in long-term ALI cultures. First, increased vimentin contents were observed in early, short-term, and mid-term, but not long-term, COPD3-4 versus NS cultures and in early Smo- and COPD1-2 cultures (Fig 5A and B). Accordingly, COPD and Smo-AE displayed increased fibronectin release up to mid-term cultures versus NS (Fig 5C). No difference remained in long-term cultures, whereas no difference was observed regarding vimentin

mRNA abundance (*VIM*, Fig S4). Second, four EMT-related transcription factors (*SNAI1*, *SNAI2*, *TWIST1*, and *ZEB*) were assessed for mRNA abundance. Similar patterns to vimentin and fibronectin were observed, showing increased levels in early, short-term, and mid-term ALI-AE in COPD samples that progressively diminished in long-term cultures (Fig 5D). This increase was also noted in smokers' ALI-AE, although with less consistency. In conclusion, these data demonstrate that EMT features gradually fade away in COPD.

### Polarity-related pIgR/SC expression

Smo- and COPD-derived AE released less apical SC than NS, from short- up to long-term ALI cultures (Fig 6A), a decrease that was more prominent in COPD3-4 patients and correlated with disease severity (Fig 6B). Concordantly, COPD3-4 AE displayed decreased *PIGR* mRNA abundance throughout the long-term cultures. Although this was not significant on isolated time points (Fig 6C), longitudinal data demonstrated decreased *PIGR* mRNA abundance in COPD3-4 AE versus other groups (Fig 6C, right panel). In addition,

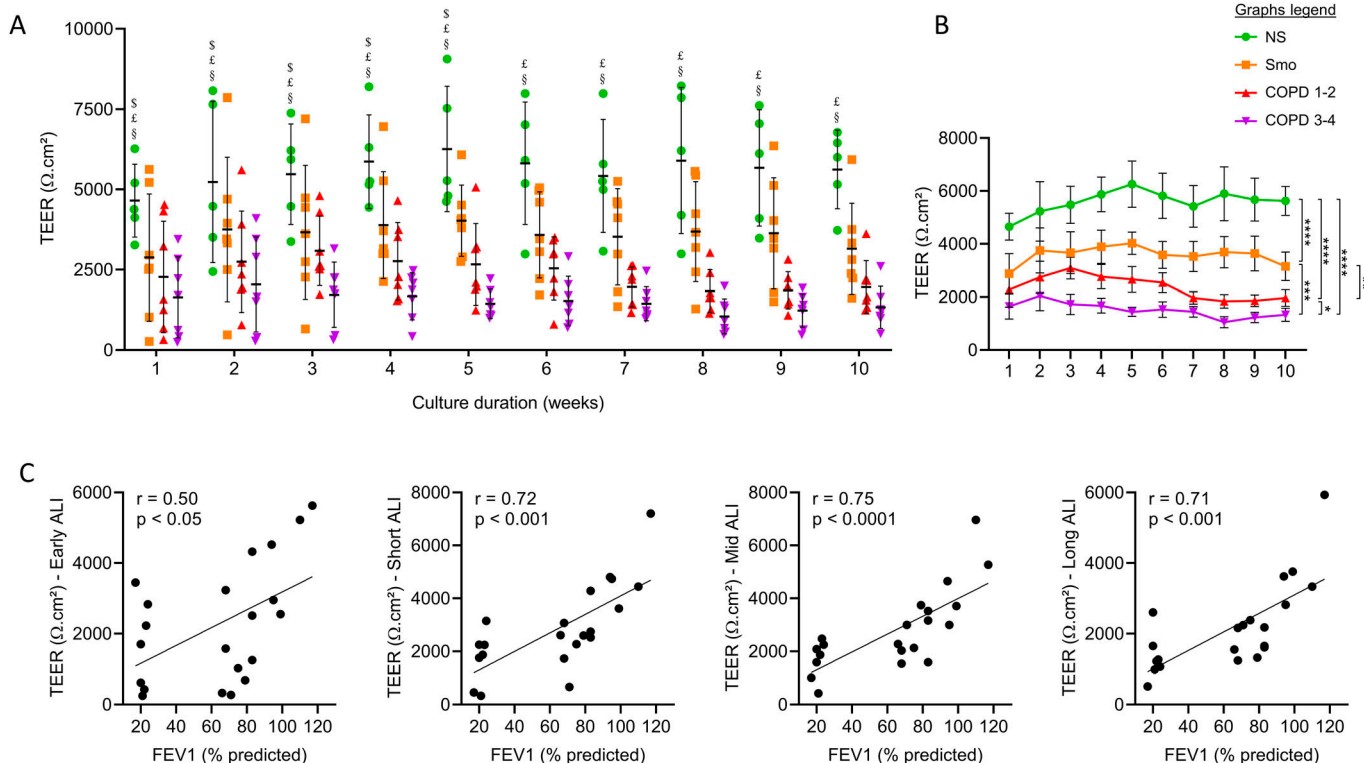

**Figure 1. COPD and Smo AE displays persistently decreased TEER compared with NS AE.**
**(A)** TEER in the ALI-AE from NS, Smo, and COPD patients (GOLD classification from one to four according to the spirometric severity of the disease). ALI-AE derived from COPD patients display a sharp decrease in TEER as compared with (non)-smokers at all time periods, that is also observed to a lesser extent in mild and moderate COPD. $, £, and § highlight significant decreases in Smo, COPD1-2, and COPD3-4 patients, respectively, as compared with NS. **(B)** Longitudinal analysis of the evolution of the TEER in the ALI-AE from NS, Smo, mild-to-moderate COPD, and (very) severe COPD patients, showing a smoking-related persistent barrier dysfunction that is further enhanced in COPD. **(C)** The barrier dysfunction observed in COPD, witnessed by the TEER decrease, significantly correlates with the disease severity assessed by the FEV1, from early cultures up to long-term cultures. Graphs include only Smo and COPD samples to specifically assess the correlation with the disease severity. ***, **** indicate *P*-values of less than 0.001 and 0.0001, respectively. Bars indicate mean ± SD. AE, airway epithelium; ALI, air-liquid interface; COPD, chronic obstructive pulmonary disease; FEV1, forced expired volume in 1 s; NS, non-smokers; SEM, standard error of the mean; Smo, smokers; TEER, transepithelial electric resistance.

considering that epithelial differentiation in ALI is completed at 5 wk, the acquisition of maximal pIgR protein levels was delayed in COPD AE (Fig 6D). Concordantly, decreased pIgR levels were also observed in situ in bronchial sections from Smo and COPD AE (Fig 6E).

These data show that both pIgR expression and functionality (SC release) are impaired in COPD. Along with impaired AJCs (see above), this indicates that the polarity of the AE in COPD is persistently altered.

### Inflammatory cytokine production by the COPD AE

IL-8/CXCL-8 and IL-6 are two main epithelial pro-inflammatory cytokines and were assayed in ALI-AE. A trend for increased *CXCL8* mRNA abundance was seen in Smo and COPD, in early to mid-term culture (Fig S5A). Accordingly, IL-8/CXCL-8 release was strongly increased up to mid-term cultures from Smo- and COPD-derived versus NS, whereas this difference disappeared afterwards (Fig S5A). Although no difference was observed regarding *IL6* mRNA abundance (Fig S5B), IL-6 production was increased in Smo- and

COPD-AE. In contrast to IL-8/CXCL-8, IL-6 up-regulation persisted in Smo and COPD long-term cultures (Fig S5C).

### Exogenous inflammation induces COPD-like AE features

Whether exogenous inflammation could promote COPD-like changes in the AE was assessed by supplementing the culture medium with IL-6, TNF-$\alpha$, and IL-1$\beta$ (all at 5 ng/ml) for up to 5 wk. First, TEER was decreased in inflammatory conditions in early cultures with this defect persisting in subsequent weeks (Fig 7A). Cytokine-exposed cultures displayed similar values irrespectively of the original phenotype, possibly indicating a maximal effect on barrier (dys)function at the used concentrations. Second, cultures exposed to inflammatory cytokines overexpressed EMT-related proteins, with increased fibronectin release and vimentin contents (Fig 7B and C). Interestingly, this induction that was not significant in NS was strikingly increased in COPD compared to pooled controls. These results show that the COPD AE, whereas losing its intrinsic EMT features, remains prone to develop EMT upon inflammatory exposure. Third, cytokine-induced inflammation altered epithelial polarity, with SC apical release being

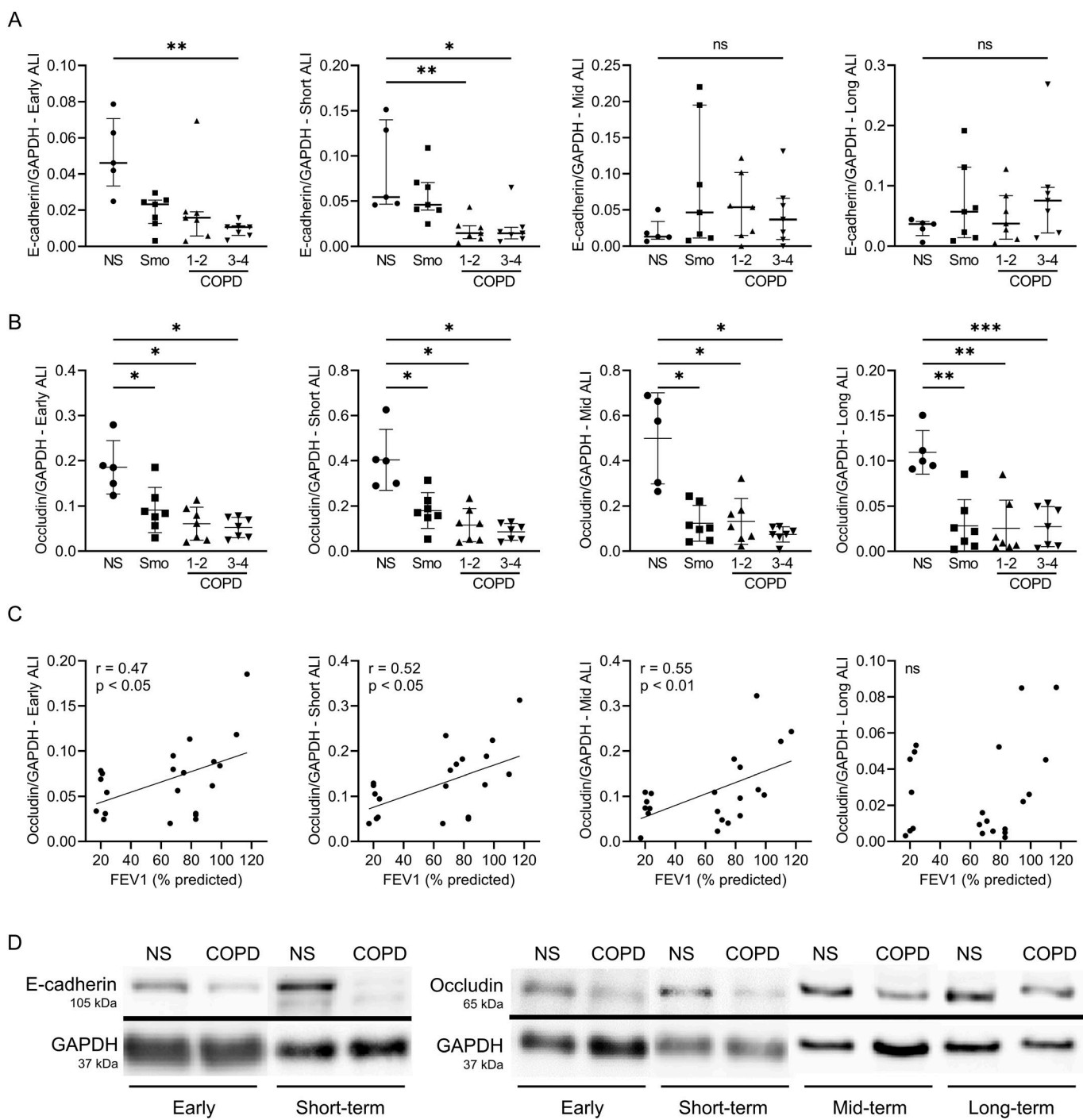

**Figure 2.  COPD and Smo AE displays decreased E-cadherin and occludin protein expression compared with NS.**
**(A)** Decreased E-cadherin protein levels in early and short-term ALI-AE, but no more at mid-term and long-term, in Smo and COPD ALI-AE as compared with that of NS. **(B)** Decreased occludin protein levels, observable from early up to long-term cultures, in Smo and COPD ALI-AE as compared with that of NS. **(C)** Occludin decrease observed in COPD, significantly correlates with the disease severity assessed by the FEV1, from early cultures up to mid-term (but not long-term) cultures. Graphs include only Smo and COPD samples to specifically assess the correlation with the disease severity. **(D)** Representative blots for E-cadherin and occludin in NS and very severe COPD AE, showing decrease in E-cadherin expression in early and short-term ALI and decrease in occludin expression in COPD ALI-AE from early up to long-term cultures. *, **, *** indicate *P*-values of less than 0.05, 0.01, and 0.001, respectively (analyzed using the Kruskal-Wallis test followed by Dunn's post hoc test). Bars indicate median ± interquartile range. AE, airway epithelium; ALI, air-liquid interface; COPD, chronic obstructive pulmonary disease; FEV1, forced expired volume in 1 s; NS, non-smokers; ns, not significant; SEM, standard error of the mean; Smo, smokers.

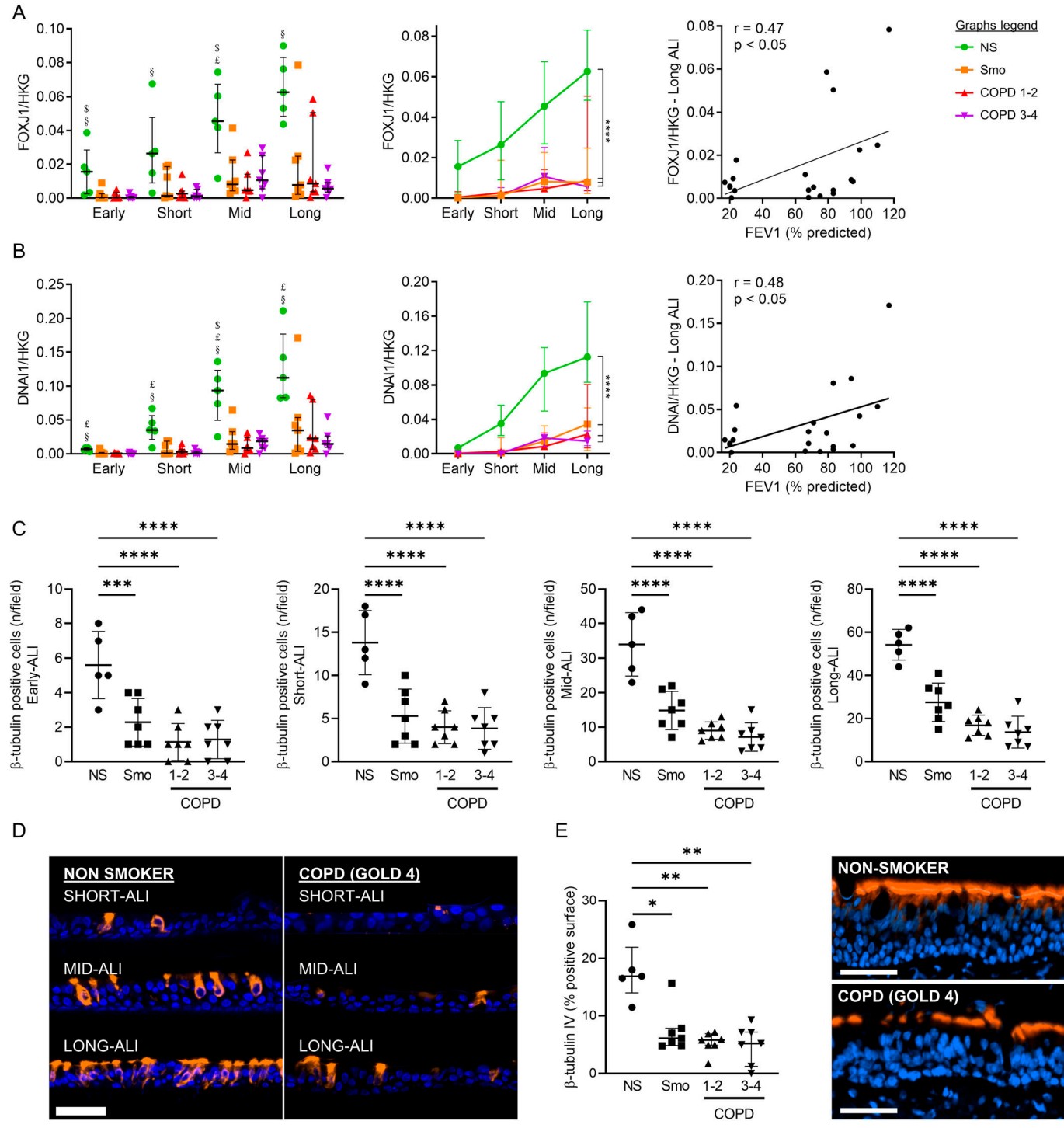

**Figure 3. Altered differentiation programming towards ciliated cells in Smo and COPD patients.**
**(A)** Decreased *FOXJ1* expression in Smo and COPD AE, from early up to long-term ALI culture. Longitudinal analysis (middle graph) shows a strong, persistent downregulation in Smo and COPD, as compared with NS. At some time periods (as represented here in long-term ALI-AE), *FOXJ1* expression correlated with the severity of the disease witnessed by the FEV1 (left graph). Correlation graph includes only Smo and COPD samples to specifically assess the correlation with the disease severity. $, £, and § highlight significant decreases in Smo, COPD1-2, and COPD3-4 patients, respectively, as compared with NS. **(B)** Decreased *DNAI1* expression in Smo and COPD AE, from early up to long-term ALI culture. Longitudinal analysis (middle graph) shows a strong, persistent down-regulation in Smo and COPD, as compared with NS. At some time periods (as represented here in long-term ALI-AE), *DNAI1* expression correlated with the severity of the disease witnessed by the FEV1 (left graph). Correlation graph includes only Smo and COPD samples to specifically assess the correlation with the disease severity. **(C)** Decreased numbers of bêta-tubulin IV–positive ciliated cells in Smo and COPD-derived ALI at every time-period. **(D)** Illustrative TSA-enhanced immunofluorescence staining for bêta-tubulin IV, demonstrating the differentiated emergence of ciliated cells in NS and COPD-derived ALI cultures. Scale bar, 50 μm. **(E)** Quantification of bêta-tubulin IV + surface in bronchial sections demonstrating reduced in situ expression of bêta-tubulin IV in native Smo and COPD AE, as illustrated by TSA-enhanced immunofluorescence staining for bêta-tubulin IV (right picture; scale bar, 50 μm). *, **, ***, **** indicate *P*-values of less than 0.05,

decreased in all cultures except in COPD3-4, probably because of pre-existent pIgR/SC severe impairment in this group (Fig 7D).

In conclusion, exogenous inflammation induces COPD-like changes including barrier dysfunction, EMT, and altered polarity in NS- and Smo-AE. In addition, the COPD-derived AE remained prone to develop EMT upon inflammation.

## Discussion

Exploiting unprecedented long-term cultures of ALI-reconstituted human AE from COPD and controls, we demonstrate that the COPD AE retains several abnormalities associated with the disease for prolonged periods of time and reveal key temporal relationships between smoking history and airway cell imprinting as assessed in ALI cultures. We also show that inflammation may recapitulate a COPD-like phenotype and that COPD cells are prone to EMT programming upon inflammatory stimulation.

COPD is a chronic and progressive disease because of repeated injury of the airway epithelial-mesenchymal unit by toxics, leading to AE activation and remodeling. COPD patients who quit smoking may benefit from decreases in mortality, respiratory symptoms, and lung function decline (Fletcher & Peto, 1977; Anthonisen et al, 1994; Willemse et al, 2004; Bai et al, 2017) and display reduced AE remodeling and goblet cell hyperplasia when smoking cessation exceeds 3.5 yr (Lapperre et al, 2007). In contrast, no change was observed following smoking cessation regarding axonemal abnormalities in ciliated cells (Verra et al, 1995), airway mucosal inflammation (Gamble et al, 2007), sputum IL-8/CXCL-8, or neutrophils (Willemse et al, 2005; Louhelainen et al, 2009), suggesting permanent alterations in these compartments.

This study explores whether the irreversible nature of the disease is imprinted in the AE in such a way that aberrant features of the native epithelium persist in long-term cultures, independently of signals provided by repeated insults and/or mesenchymal cells and surrounding leukocytes. To study this, we chose to use the ALI model as it was previously shown to recapitulate, at least to some extent, the native COPD phenotype (Gohy et al, 2014, 2015, 2019; Carlier et al, 2018; Carlier et al, 2020), and we prolonged the culture up to 10 wk, an unusually long and so far unreported duration in this model.

A major abnormality that persists in the COPD AE is the barrier and junctional defect (Carlier et al, 2021) that was initiated in Smo and was associated with decreased protein levels of E-cadherin and occludin (Figs 1 and 2). This observation corroborates previous studies showing that CS exposure disrupts AJCs (Shaykhiev et al, 2011). We here show that this alteration further persists in long-term cultures for occludin. In addition, changes in TEER and E-cadherin/occludin expression are further aggravated according to the presence and severity of COPD. No difference was observed regarding mRNA abundance for the main AJCs' components, suggesting post-transcriptional regulation.

In line with the requirement of AJCs' integrity to ensure baso-apical epithelial polarity (Cereijido et al, 1998), we show that AE polarity is persistently impaired in COPD. Aside from above-mentioned AJC disruption, the pIgR/SC system, which allows baso-apical transcytosis of polymeric immunoglobulins (Carlier et al, 2016), was also shown to be defective in COPD (Gohy et al, 2014). Our study demonstrates that this impairment persists over time, with decreased SC apical release and *PIGR* mRNA abundance in COPD3-4 AE (Fig 6). Interestingly, those data are contrasting with previous findings in asthma where pIgR down-regulation, although being similarly present in situ, does not persist in ALI (Ladjemi et al, 2018), suggesting distinct mechanisms driving epithelium pathology in asthma and COPD.

Ciliated cell hypoplasia — with decreased *FOXJ1* and *DNAI1* mRNA abundance and reduced bêta-tubulin IV⁺ cell numbers — also persists in long-term COPD-derived cultures, matching in situ findings (Figs 3 and S3). In addition, goblet cell hyperplasia persists in Smo, as *SPDEF* and *FOXA3* expression, along with MUC5AC⁺ cell numbers, remain higher in active smokers and ex-smokers who quit for less than 4 yr, compared with long-quitters (Fig 4). These results support in situ data and corroborate previous findings indicating that the muco-secretory trait relates more directly to smoking rather than to COPD (Lapperre et al, 2007).

In contrast, some abnormalities are observed solely in short- to midterm cultures, such as EMT. EMT is a physiological process notably involved in tissue repair (type II EMT), which is dysregulated in COPD, where it is thought to contribute to airway fibrosis (Bartis et al, 2014). In addition, EMT seems to play a role in the increased susceptibility to lung cancer in COPD individuals; however, further confirmation is required (Mahmood et al, 2021). In this study, we show aberrant EMT features in Smo and COPD AE during the first weeks of culture, with increases in both vimentin contents and *SNAI1*, *SNAI2*, *TWIST1*, and *ZEB* mRNA levels. However, these features vanish from mid-term cultures onwards, with complete disappearance occurring earlier in Smo than in COPD (Fig 5). Nevertheless, the COPD AE remains prone to reactivate EMT programming upon inflammatory condition, suggesting the existence of imprinting of the COPD AE by previous (in vivo) exposures conditioning its responses to further stimulation (Fig 7).

As observed for EMT, up-regulated IL-8/CXCL-8 release by Smo and COPD AE also faded away from mid-term cultures onwards, whereas IL-6 overproduction persisted in long-term ALI-AE from both Smo and COPD patients (Fig S5). In vitro exposure to inflammatory cytokines reproduced or aggravated alterations in barrier and polarity features and EMT in controls and COPD AE, respectively.

The fundamental mechanisms of these observations question the nature of epithelial memory. Inflammatory memory refers to memories of previous immune events enabling barrier tissues to rapidly recall distinct environmental exposures, which may be stored not only in immune cells but also in epithelial and mesenchymal cells (Naik et al, 2017; Ordovas-Montanes et al, 2020). Whereas memory classically refers in the immune system to

---

0.01, 0.001, and 0.0001, respectively. Bars indicate median ± interquartile range (A, B, E) or mean ± SD (C). AE, airway epithelium; ALI, air-liquid interface; COPD, chronic obstructive pulmonary disease; CT, control; FEV1, forced expired volume in 1 s; HKG, housekeeping genes; NS, non-smokers; SEM, standard error of the mean; Smo, smokers; y, years.

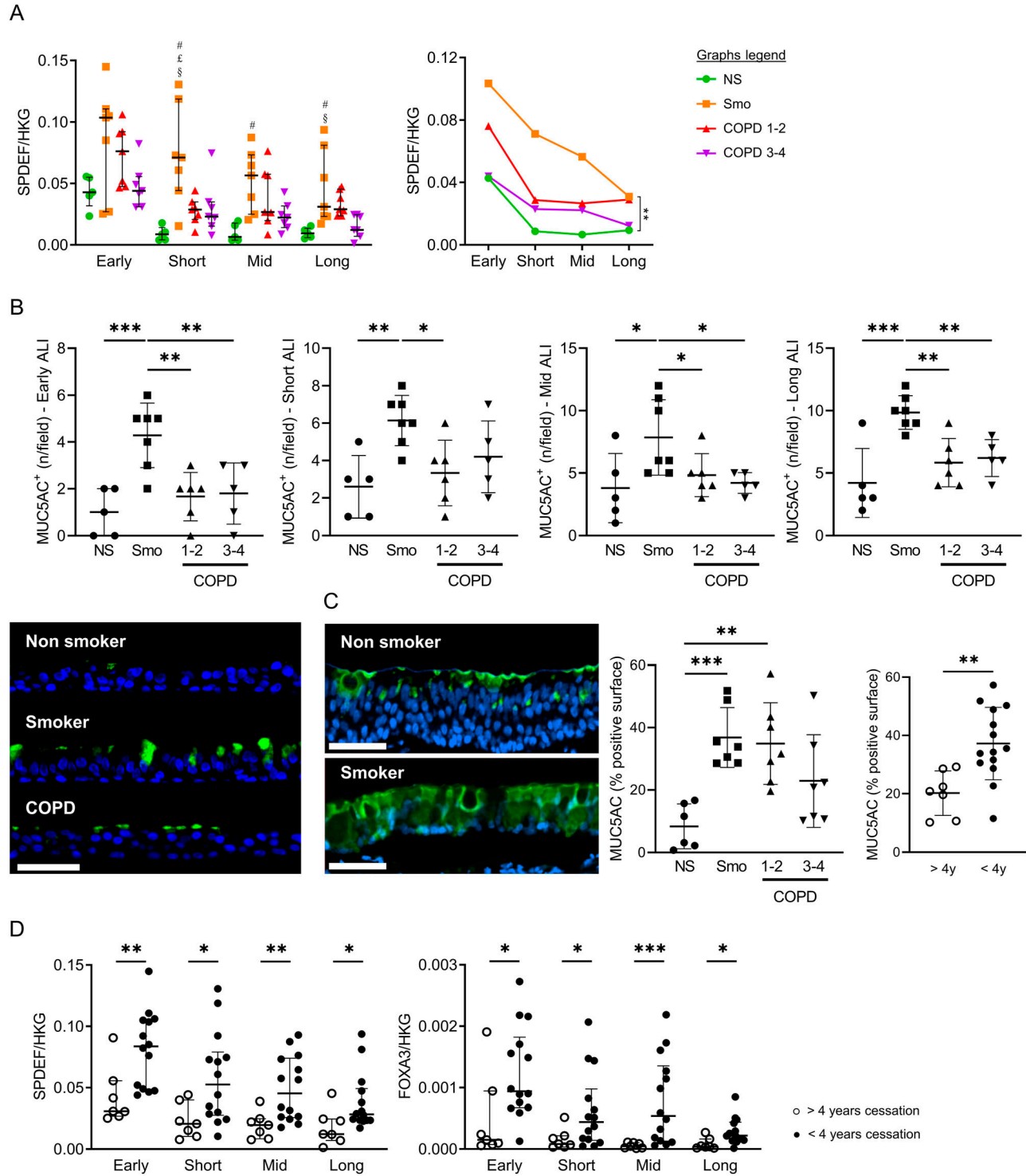

**Figure 4. Altered differentiation programming towards goblet cells in Smo and COPD patients.**
**(A)** Increased *SPDEF* expression in Smo AE as compared with NS in short-term, mid-term, and long-term ALI culture, as depicted for each timepoint (left graph) and in a longitudinal analysis (right graph, significance bar: Smo versus NS). Of note, error bars were not added on the longitudinal data for reasons of readability. #, £, and § highlight significant decreases in NS, COPD1-2, and COPD3-4 patients, respectively, as compared with NS. **(B)** Increased MUC5AC-positive goblet cells in Smo AE as compared with NS and COPD patients in early, short-term, mid-term, and long-term ALI culture. Bottom left picture: illustrative TSA immunofluorescence staining for MUC5AC, showing increased positive goblet cells in ALI cultures (here, in mid-term ALI-AE) from Smo as compared with NS and COPD. **(C)** In situ, in bronchial surgical sections, MUC5AC-positive surface is increased in Smo AE as compared with NS and COPD. In addition, MUC5A positive surface is significantly higher in patients who quit smoking for less than 4 yr (plain dots) as compared with long-quitters (hollow dots). **(D)** At every time period, *SPDEF* (left graph) and *FOXA3* (right graph) mRNA levels are decreased in ALI-AE from Smo who quit smoking for more than 4 yr (plain dots) as compared with long quitters (hollow dots). *, **, *** indicate *P*-values of less than 0.05,

somatic mutations underlying adaptive antibody responses to recall antigens, "inflammatory memory" is less well defined and may include epigenetic modifications and chromatin changes that may drive persistent alterations in damaged tissues. In the airways, progenitor basal cells are prime candidates to retain epithelial memory, as do epithelium progenitors in the skin towards inflammatory or mechanical stress (Naik et al, 2017) and in the gut towards dietary components (Beyaz et al, 2016). Interestingly, WNT signalling, which is up-regulated in the COPD AE (Carlier et al, 2020), has been shown to be involved in the latter form of epithelial memory and may regulate stemness and tumorigenicity. In line with the recent hypothesis that basal cells serve as repositories of allergic inflammatory memory in airway epithelial cells (Ordovas-Montanes et al, 2020), one could propose that airway stem cells also store the memory of repeated previous injuries by inhaled toxics such as cigarette smoke.

Our study has several limitations. First, an effect of treatments (e.g., inhaled corticosteroids in severe patients) on the findings cannot be excluded. However, in vitro studies on intestinal cell lines and primary HBEC cultured at the ALI showed that dexamethasone induced increased TEER, claudin-2 (Fischer et al, 2014), and E-cadherin (Carayol et al, 2002a, 2002b) expression, whereas budesonide exposure of ALI-HBEC counteracted CS-induced barrier dysfunction (Heijink et al, 2012). In addition, dexamethasone and fluticasone propionate improved TGF-$\beta_1$-induced EMT in A549 cells (Yang et al, 2017), suggesting that corticosteroids may rather improve barrier function and EMT changes. Second, airway progenitor (basal) cells are suspected to display reduced self-renewal (Ghosh et al, 2018), although conflicting data exist (Staudt et al, 2014). One cannot exclude that COPD ALI-AE could originate from basal cells with lower replication rate. However, no difference was observed regarding time to reach confluence (both in flasks and in inserts) across study groups. In addition, one could expect a "catch-up" effect in long-term cultures, which was not the case, arguing for a progenitor imprinting in COPD, as already suggested by Ghosh and colleagues (Ghosh et al, 2018). Third, initial samples were either collected from patients undergoing lobectomy (NS, Smo, and COPD1-2 groups, in most cases for oncologic indication) or from explants issued from patients undergoing lung transplantation (and therefore free from cancer, COPD3-4 group). Although the samples were collected as far as possible from the tumor, an effect of distant cancer on the results cannot be ruled out. However, the results most likely to be influenced by a cancer (EMT, barrier dysfunction) are the most altered in the COPD3-4 group, free of cancer. On the other hand, the observed alterations in the COPD3-4 group are already present to a lesser extent in the COPD1-2 group, making an effect related to group disparity highly unlikely. Finally, the imprinting observed in this model of ex vivo reconstituted epithelium that implies proliferation and differentiation of basal cells in prolonged ALI culture, could be confirmed in other models such as 3D airway organoids.

In conclusion, this study demonstrates that the AE from Smo and COPD stores the memory of its native state and previous insults from cigarette smoking. This memory is multidimensional, including alterations in barrier function, epithelial polarity, lineage differentiation, IL-6 release, and EMT reprogramming. Further studies are required to explore the mechanisms of this memory, confirm the exact cellular niche retaining it, and identify putative targets for future therapies.

# Materials and Methods

### Study population and lung tissue samples

Lung surgical tissue from lobectomies was obtained from both non-smokers and smokers controls, and from mild/moderate COPD patients undergoing lung surgery for a solitary tumor, whereas lung explants were obtained to analyze lung tissue from patients with (very) severe COPD. Subjects were sorted on basis of clinical diagnosis and pulmonary function tests, according to the GOLD 2001 classification (Pauwels et al, 2001). Patients with any other lung disease than COPD (e.g., asthma, lung fibrosis) were excluded from the study. All patients received information and signed a written consent to the study protocol, which was approved by the Local Clinical Ethical Committee (reference 2007/19MARS/58 for UCLouvain, S52174, and S55877 for KULeuven).

A primary proximal AE was reconstituted in vitro from all subjects and subjected to long-term culture (10 wk, n = 26) or mid-term culture with cytokine stimulation (5 wk, n = 25). Tables 1 and S1 recapitulate patients' characteristics for each population.

### In vitro reconstitution of primary human AE on ALI culture

ALI-cultures were conducted as previously published (Carlier et al, 2020). A large piece of lobar or segmentar bronchus (third or fourth generation) was selected from lobectomies or explants, located as far as possible from the tumor site (in the case of lobectomies) and submitted to pronase digestion overnight at 4°C, to derive primary HBEC for each sample. HBEC were seeded in 75 cm$^2$ flasks (2 × 10$^6$ alive cells/flask), then cultured in retinoic acid-supplemented Bronchial Epithelial Cell Growth Basal Medium (BEBM; Lonza) until confluence, which was reached at day 7 (±2 d [SD]), with no difference across the groups. Confluent cells were exclusively (>99.9%) p63$^+$ basal cells. Cells were then detached (passage 1), assessed for viability, and seeded at a density of 80,000 alive cells/well on 24-well polyester filter-type inserts (0.4-$\mu$m pore size; Corning) coated with 0.2 mg/ml collagen IV (Sigma-Aldrich) until a confluent monolayer was obtained. Of note, mean cell viability before seeding in inserts was 92.4% (±3.4% [SD]), and mean time to confluence in submerged conditions was 6 d (±0.8 d [SD]), with no difference between the study groups. The culture was then carried out in ALI.

26 samples were carried out for 10 wk and assessed for several epithelial properties. Twenty-five samples were carried out for 5 wk

---

0.01, 0.001, and 0.0001. Bars indicate means ± SD (A, B, C) or medians ± interquartile ranges (D). AE, airway epithelium; ALI, air-liquid interface; COPD, chronic obstructive pulmonary disease; CT, control; FEV1, forced expired volume in 1 s; HKG, housekeeping genes; NS, non-smokers; SEM, standard error of the mean; Smo, smokers; y, years. Scale bar, 50 $\mu$m.

**Figure 5. COPD-related EMT features vanish during long-term ALI cultures.**
**(A)** Vimentin contents are increased in early Smo cultures, and up to mid-term in COPD ALI-AE, with this increase vanishing in long-term cultures. **(B)** Illustrative gels from NS and very severe COPD-derived AE cultured at the ALI from early up to long-term, illustrating the vimentin increased contents up to mid-term in COPD. **(C)** Increased fibronectin release disappears in mid-term cultures from Smo and mild-to-moderate COPD patients, whereas it persists up to later (7 wk) in (very) severe COPD AE. **(D)** Increased mRNA levels of *SNAI1*, *SNAI2*, *TWIST1*, and *ZEB* globally disappear in mid-term cultures from Smo and COPD patients, with punctual persistence of SNAI2 increase in long-term Smo ALI-AE. *, ** indicate *P*-values of less than 0.05 and 0.01, respectively. Bars indicate median ± interquartile range. $, £, and § highlight significant decreases in Smo, COPD1-2, and

and used exclusively for the "inflammatory stimulation" experiment (see details below). Once in ALI, HBEC were cultured in BEBM: DMEM (1:1) medium supplemented with penicillin (100 U/ml), streptomycin (100 $\mu$g/ml) (Lonza), BSA (1.5 $\mu$g/ml), retinoic acid (30 ng/ml) (Sigma-Aldrich), and BEGM SingleQuotsTM Supplements and Growth Factors (Lonza), including bovine pituitary extract (52 $\mu$g/ml), insulin (5 $\mu$g/ml), hydrocortisone (0.5 g/ml), transferrin (10 $\mu$g/ml), epinephrine (0.5 $\mu$g/ml), epidermal growth factor (0.5 ng/ml), and triiodothyronine (3.25 ng/ml). No culture infection occurred among the 51 samples.

Every week during ALI culture, basolateral media were collected, and the apical pole of HBEC was washed with 300 $\mu$l sterile PBS before centrifugation for 5 min at 10,000$g$. Transwell inserts were fixed by direct immersion in 4% buffered formaldehyde, before incubation in PBS (pH 7.4) and embedding in paraffin blocks. ALI-HBEC were also processed for mRNA abundance or Western blot analyses (see below).

The 25 samples that underwent 5 wk ALI culture were exposed (48 h) or not (control condition) to a pro-inflammatory cytokine cocktail including IL-1$\beta$, IL-6, TNF-$\alpha$, each at 5 ng/ml (Miltenyi Biotec) in the basolateral compartment following a preliminary titration experiment (10, 5, and 2,5 ng/ml), where no cytotoxicity (release of lactate dehydrogenase < 5%) was shown at 5 ng/ml (data not shown).

TEER was assessed every week for each sample, using the EMD Millipore Millicell-ERS Volt-Ohmmeter (Thermo Fisher Scientific) after transiently filling the apical pole with sterile PBS and correcting for the resistance of the transwell membrane.

## Quantitative reverse transcription-polymerase chain reaction (qRT-PCR)

RNA extraction, reverse transcription, and qRT-PCR were performed as previously described (Bustin et al, 2013). Total RNA was extracted from reconstituted ALI-cultured epithelia using TRIzol reagent (Thermo Fisher Scientific). 500 ng of RNA was reverse-transcribed with RevertAid H minus reverse transcriptase kit with 0.3 $\mu$g of random hexamer, 20 U of RNase inhibitor, and 1 mM of each dNTP (Thermo Fisher Scientific) following the manufacturer's protocol in a thermocycler (Applied Biosystems). The expression levels were quantified by real-time quantitative PCR with the CFX96 PCR (Bio-Rad). The reaction mix contained 2.5 $\mu$l of complementary desoxyribonucleic acid diluted 10-fold, 200 nM of each primer (primers properties are detailed in Table S2), and 2x iTaq UniverSybr Green Supermix (Bio-Rad) in a final volume of 20 $\mu$l. The cycling conditions were 95°C for 3 min followed by 40 cycles of 95°C for 5 s and 60°C for 30 s. To control the specificity of the amplification products, a melting curve analysis was performed. The copy number was calculated from the standard curve. Data analysis was performed using Bio-Rad CFX software (Bio-Rad). Expression levels of target genes were normalized to the geometric mean of the values of three housekeeping genes (*RPL27*, *RPS13*, *RPS18*).

## Western blot assays

Cells were lysed with 150 $\mu$l of Laemmli's sample buffer containing 0.7 M 2-mercaptoethanol (Sigma-Aldrich) and lysates were stored at −20°C. After thawing, samples were heated at 100°C for 5 min, loaded in a SDS–PAGE gel before migration at 100 V for 15 min and then at 180 V for 50 min. Cell proteins were transferred onto a nitrocellulose membrane (Thermo Fisher Scientific) at 0.3 A for 2 h 10 min at RT. The membranes were blocked with 5% wt/vol BSA (Sigma-Aldrich) in Tris-buffered saline with 0.1% Tween 20 (Sigma-Aldrich) for 1 h at RT, then washed and incubated overnight at 4°C with a primary antibody according to the target protein (see Table S3 listing used primary and secondary antibodies). Membranes were then incubated for 1 h at RT with HRP-conjugated secondary anti-rabbit (Cell Signalling) or anti-mouse (Sigma-Aldrich) IgG.

Revelation was performed by chemiluminescence (GE Healthcare) before detection by Chemidoc XRS apparatus (Bio-Rad) and quantification by Quantity One software (Bio-Rad).

## ELISA for SC, IL-8/CXCL-8, IL-6, and fibronectin

Levels of SC (in apical washes) and cytokines (in basolateral supernatants) were measured by sandwich ELISA, as previously described (Pilette et al, 2001; Pilette et al, 2003). Basolateral IL-8/CXCL8 and IL-6 release were assessed by sandwich ELISA, following manufacturer's instructions (of IL-6). Briefly, 96-well plates were coated overnight, at 4°C, with anti-IL-8/CXCL8, -IL-6, and -SC antibodies diluted in bicarbonate buffer (pH 9.6). Then, after blocking with 1% wt/vol BSA in phosphate-buffered saline for 90 min at 37°C, HBEC apical washes (for SC) or basolateral supernatants (for IL-8/CXCL-8 and IL-6) were incubated for 60 min at 37°C, along with standard samples. Detection was performed with a first incubation with the corresponding biotinylated antibody (anti-fibronectin, -SC, -IL-8/CXCL-8 or -IL-6), followed by a second incubation with HRP-linked anti-mouse IgG, for 1 h each. Revelation was performed with 3,3′,5,5′-tetramethylbenzidine (TMB; Thermo Fisher Scientific) and stopped with $H_2SO_4$ 1.8 M.

Fibronectin release in the basal medium was assessed by direct ELISA, as previously reported (Gohy et al, 2015; Collin et al, 2020). HBEC basolateral washes and fibronectin standard were coated in plates with bicarbonate buffer overnight (pH 9.6), at 4°C. After blocking with 1% wt/vol BSA in phosphate-buffered saline for 90 min at 37°C, detection was performed with a first incubation with mouse anti-fibronectin, followed by a second incubation with HRP-linked anti-mouse IgG, for 1 h each. Revelation was performed with TMB and stopped with $H_2SO_4$ 1.8 M.

## Immunofluorescence staining using tyramide signal amplification

Five micron-sections of bronchial tissue or reconstituted ALI epithelium, fixed in 4% formaldehyde and paraffin-embedded, were deparaffinised in toluene and rehydrated through a graded series from ethanol to water. Antigen retrieval was performed in citrate

---

COPD3-4 patients, respectively, as compared with NS. AE, airway epithelium; ALI, air-liquid interface; COPD, chronic obstructive pulmonary disease; GAPDH, glyceraldehyde-3-phosphate dehydrogenase; NS, non-smokers; ns, not significant; Smo, smokers; w, weeks.

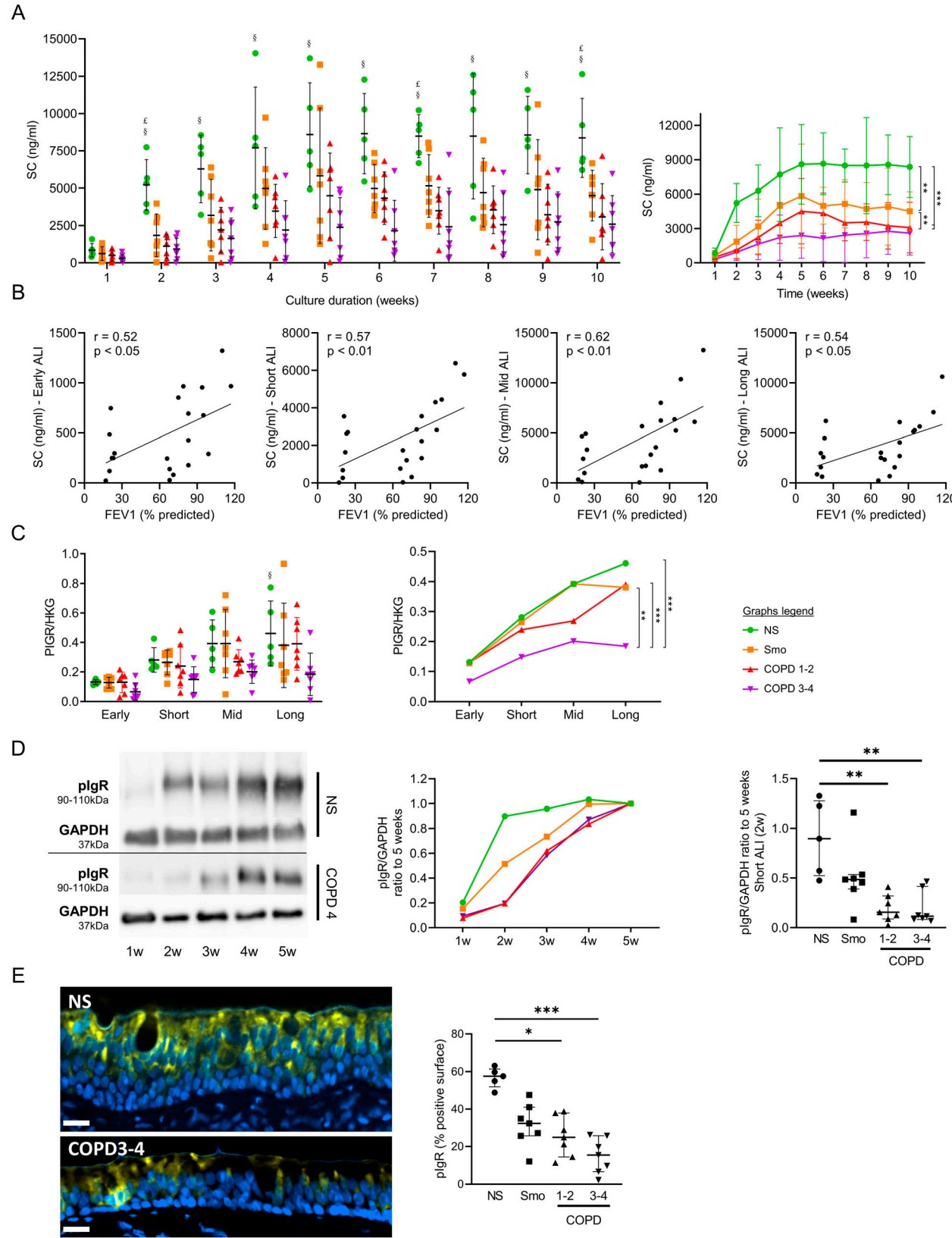

**Figure 6.   Impaired polarity, witnessed by a disruption and a delayed acquisition of PIGR/SC expression, is imprinted in the COPD AE.**
**(A)** SC apical release in ALI-AE from Smo and COPD patients is decreased as compared with NS, from short-term ALI-AE up to long-term ALI-AE. $, £, and § highlight significant decreases in Smo, COPD1-2, and COPD3-4 patients, respectively, as compared with NS. **(B)** SC decreased release in the COPD AE moderately correlates with the

buffer (pH 6.0 containing 0.1% of triton) using a pressure cooker at 15 PSI for 5 min. Sections were blocked for non-specific antigen binding by incubation in Bloxall (Vector Laboratories Inc.) for 15 min and then in 0.3% hydrogen peroxide with 5% goat serum (Bio-Rad) for 30 min. Staining first included a 30-min protein blocking with 5% goat serum, then the primary antibodies diluted in 5% normal goat serum solution were applied, then the appropriate SuperBoost goat anti-rabbit or anti-mouse, poly-HRP-conjugated secondary antibody (Thermo Fisher Scientific) was applied for 40 min. HRP-conjugated polymer mediated the focal covalent binding of a fluorophore using tyramide signal amplification. In surgical samples, this cycle was repeated twice to allow multiplex staining of pIgR, bêta-tubulin IV, and MUC5AC. Table S4 recapitulates the antibodies and fluorophores that were used. Finally, sections were counterstained with Hoechst (Thermo Fisher Scientific) diluted at 10 µg/ml in TBS-BSA 5% and mounted with Dako fluorescence mounting medium (Dako). For negative controls, we used rabbit or mouse isotype controls at the same concentration as the corresponding primary antibodies (diluted in 5% normal goat serum).

### Immunofluorescence staining quantification

For ALI-reconstitued AE, bêta-tubulin IV⁺ and MUC5AC⁺ cells were manually counted at different time points (1 wk, early; 2 wk, short-term; 4 wk, mid-term; and 9 wk, long-term). For each sample, five fields (20x magnification) were analyzed, and the arithmetic mean of the five fields was calculated.

For in situ bronchial sections, QuPath 0.4.2 analysis tool (Bankhead et al, 2017) was used. First, epithelial layers were manually delineated on each slide. Staining thresholds for MUC5AC and bêta-tubulin IV staining were defined before software analysis. Finally, the positive surface was calculated within the total area delineated and expressed as a percentage.

### Statistical analysis

Before analysis, all data were assessed for normality using Shapiro-Wilk and Kolmogorov-Smirnov tests. Parametric tests were used only when all groups were considered normal with the two tests. Data were expressed as means and SD for data reaching normality, whereas non-normally distributed data were expressed as medians and interquartile ranges. Data were analyzed with JMP Pro, Version 14 (SAS Institute Inc.) and GraphPad Prism version 8.0.2 for Windows

(GraphPad Software). $P$-values < 0.05 were considered statistically significant.

For timepoint analyses, Brown-Forsythe and Welch ANOVA tests, followed by Holm-Sidak's multiple comparisons tests (each experimental group versus non-smoker controls), were used for normally distributed data (Figs 1A, 2B, 3A and C, 4B and C, and 6A, C and E), whereas Kruskal-Wallis tests, followed by Dunn's multiple comparisons tests (each experimental group versus non-smoker controls), were used for non-normally used data (Figs 2A, 3B and E, 4A, 5A, C and D, and 6D).

For two-group comparisons, unpaired t-tests were used for normally distributed data (Fig 4C [right graph]), whereas Mann-Whitney U tests were used for non-normally distributed data (Fig 4D). For paired data (Fig 7), paired Wilcoxon signed rank tests were used (Fig 7B and C). All correlations were assessed by using linear regression and Cohen's κ coefficient calculation.

For longitudinal analyses (Figs 1B and 3A and B [middle graphs], Fig 4A [right graph], Fig 5C [right graph], Fig 6A and C [right graphs], Fig 7A and B [right graphs]), linear mixed models were built using JMP Pro, Version 14. The models aimed at examining the effects of time (weeks of culture) and phenotype of the sample on continuous variables, and were designed to take into account repeated measures (i.e., measurements taken at different timepoints on the same samples), and specified a nesting structure where each sample was nested within a study group (i.e., NS, Smo, COPD1-2, COPD3-4). A random effect for each nested sample "sample[phenotype]" was also integrated before the model was run. Post hoc analysis with a correction of least significant difference for multiple comparisons was performed between the main effects.

## Supplementary Information

## Acknowledgements

The authors would like to thank J Van Snick for his help with IL-6 ELISA, V Lacroix, A Belhaj, Ph Eucher, and D Hoton for providing surgical tissue, A Daumerie for technical advice, M de Beukelaer and Ch Fregimilicka for their help with sample processing, and the IREC Pole of Microbiology (UCLouvain, Brussels, Belgium) for sharing their molecular biology facility. This work was supported by the Fondation Mont-Godinne, Belgium, grant to FM Carlier (N° FMG-2015-BC01, FMG-2016-BC01, and FMG-2017-BC01) and by the Fonds

---

disease severity, witnessed by the FEV1. Correlation graphs include only Smo and COPD samples to specifically assess the correlation with the disease severity. **(C)** *PIGR* mRNA abundance is not significantly decreased in (very) severe COPD ALI-AE at separate time-points (left panel), but longitudinal analysis shows a significant decrease in *PIGR* expression in (very) severe COPD AE as compared with NS, Smo, and mild-to-moderate COPD (right panel). **(D)** pIgR acquisition is delayed during the differentiation of the AE in COPD as compared with NS, as shown in the representative blots from NS and very severe COPD ALI-AE and summarized in the middle panel. The gap was maximal at 2 wk ALI culture (right panel) and catches it up only after 4 wk. **(E)** TSA immunofluorescence staining for pIgR, showing reduced positive surface in situ (i.e., in bronchial sections) in the COPD AE as compared with that of NS. *, **, *** indicate *P*-values of less than 0.05, 0.01, and 0.001, respectively (analyzed using the Kruskal-Wallis test followed by Dunn's post-hoc test, except for longitudinal analysis in (B, C), mixed model). Bars indicate mean ± SD (A, C [left], D [right], E), dots indicate mean (C [right], D [left]). AE, airway epithelium; ALI, air-liquid interface; COPD, chronic obstructive pulmonary disease; FEV1, forced expired volume in 1 s; GAPDH, glyceraldehyde-3-phosphate dehydrogenase; HKG, housekeeping genes; NS, non-smokers; ns, not significant; pIgR, polymeric immunoglobulin receptor; PV, predicted values; SC, secretory component; SD, standard deviation; Smo, smokers; w, weeks.

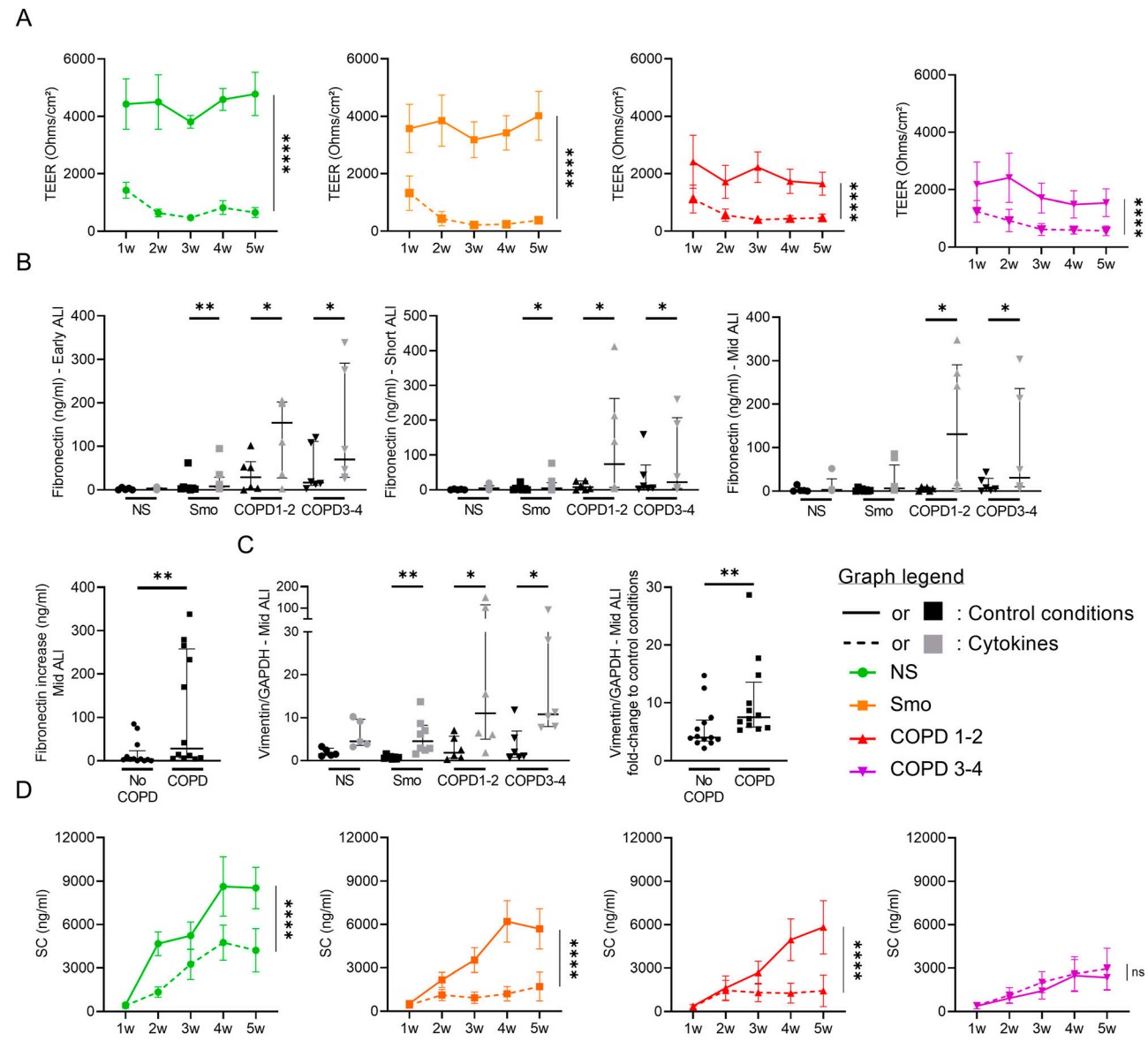

**Figure 7. Cytokine activation triggers COPD-like epithelial changes.**
**(A)** Epithelial inflammation, driven by exogenous TNF-$\alpha$, IL-1$\beta$, and IL-6, induces barrier dysfunction, witnessed by a dramatic decrease in TEER in each group (NS, Smo, COPD 1–2, COPD3-4). **(B, C)** Cytokine-induced epithelial inflammation induces EMT in Smo and COPD-derived ALI AE, witnessed by increased fibronectin release in early, short-term, and mid-term cultures (B) and vimentin expression in mid-term cultures (C). **(B, C)** The absolute increase in fibronectin (B) and vimentin (C) was significantly higher in mid-term COPD AE than in non-COPD AE (pooled Smo and NS, labeled "no COPD"). **(D)** Cytokine-induced epithelial inflammation deteriorates the epithelial polarity, witnessed by the SC apical release. No difference was seen in (very) severe COPD because of low baseline levels. *, **, **** indicate P-values of less than 0.05, 0.01, and 0.0001. Bars indicate median ± interquartile range (B, C, D) and mean ± SEM (A, E). AE, airway epithelium; ALI, air-liquid interface; COPD, chronic obstructive pulmonary disease; CT, controls; EMT, epithelial-to-mesenchymal transition; GAPDH, glyceraldehyde-3-phosphate dehydrogenase; IL, interleukin; NS, non-smokers; ns, not significant; SC, secretory component; SEM, standard error of the mean; Smo, smokers; TEER, transepithelial electric resistance; TNF-$\alpha$, tumor necrosis factor $\alpha$; w, weeks.

National de Recherche Scientifique (FNRS), Belgium, grant to FM Carlier (No 1.L505.18) and to C Pilette (No 1.R016.16 and 1.R016.18). Funders were not involved in study design, data collection, data analysis, interpretation, or writing of the manuscript.

## Author Contributions

FM Carlier: conceptualization, resources, data curation, formal analysis, supervision, funding acquisition, validation, investigation, visualization, methodology, project administration, and writing—original draft, review, and editing.

B Detry: conceptualization, data curation, and investigation.

M Lecocq: conceptualization, data curation, investigation, methodology, and writing—review and editing.

AM Collin: data curation.

T Planté-Bordeneuve: data curation and writing—review and editing.

L Gérard: conceptualization, data curation, and investigation.

SE Verleden: resources, data curation, and writing—review and editing.

M Delos: resources.

B Rondelet: resources.

W Janssens: resources.

J Ambroise: software, formal analysis, and writing—review and editing.

BM Vanaudenaerde: resources.

S Gohy: conceptualization, investigation, and writing—review and editing.

C Pilette: conceptualization, supervision, funding acquisition, validation, investigation, methodology, project administration, and writing—review and editing.

## Conflict of Interest Statement

SE Verleden reports grants to institution from Chiesi and Sanofi, outside of the submitted work. C Pilette reports grants from AstraZeneca, Chiesi, GSK, Novartis, and TEVA and consulting fees from ALK-Abello, AstraZeneca, Chiesi, GSK, Mundipharma, Novartis, Sanofi, and Staller-gènes, outside of the submitted work. FM Carlier, B Detry, M Lecocq, AM Collin, T Planté-Bordeneuve, L Gérard, M Delos, B Rondelet, W Janssens, J Ambroise, BM Vanaudenaerde, and S Gohy report no competing interests.

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
