## [Reviewer comments · Life Science Alliance]

Life Science Alliance

The memory of airway epithelium damage in smokers and COPD patients

Francois Carlier, Bruno Detry, Marylène Lecocq, Amandine Collin, Thomas Planté-Bordeneuve, Stijn Verleden, Monique Delos, Benoît Rondelet, Wim Janssens, Jérôme Ambroise, Bart Vanaudenaerde, Sophie Gohy, and Charles Pilette

DOI: <https://doi.org/10.26508/lsa.202302341>

Corresponding author(s): *Francois Carlier, Université Catholique de Louvain*

Review Timeline:	Submission Date:	2023-08-25
	Editorial Decision:	2023-10-10
	Revision Received:	2023-11-15
	Editorial Decision:	2023-12-07
	Revision Received:	2023-12-08
	Accepted:	2023-12-11

Transaction Report:

October 10, 2023

Re: Life Science Alliance manuscript #LSA-2023-02341-T

Dr. Francois Michel Carlier
Université catholique de Louvain
Institut de Recherche expérimentale et clinique
Avenue Mounier, 54
Claude Bernard Tower, 3rd floor
Brussels 1200
Belgium

Dear Dr. Carlier,

Thank you for submitting your manuscript entitled "The memory of airway epithelium damage in smokers and COPD patients" to Life Science Alliance. The manuscript was assessed by expert reviewers, whose comments are appended to this letter. We invite you to submit a revised manuscript addressing the Reviewer comments.

Thank you for this interesting contribution to Life Science Alliance. We are looking forward to receiving your revised manuscript.

Sincerely,

B. MANUSCRIPT ORGANIZATION AND FORMATTING:

Reviewer #1 (Comments to the Authors (Required)):

Many thanks for giving me the opportunity to assess this study. I am impressed by the amount of experiments and data. Finally, it seems that airway epithelial features of COPD in ALI can be more and more accurately described. There are a few issues to address

1/ I am quite convinced that the culture medium have the potential to totally change the phenotype of the airway epithelium in ALI. Accordingly, I consider mandatory explaining in details how cultures were conducted.

2/ Correlations with FEV1 are shown for TEER, occludin, foxJ1, secretory component, ... This is a bit confusing actually, given the different fields explored. This is moreover confusing that these correlations are always in the positive way. The message associated is not necessarily supporting the message defended by the authors - that I share, and I would suggest rethinking about the hypothesis behind that. Elsewhere, FEV1 actually is potentially not the marker we'd like to see as a very reliable surrogate marker in COPD.

3/ EMT is more and more clearly seen as a critical step for repair. I am not totally sure that fibronectin is really reflecting that. Maybe PCR of ZEB/TWIST can provide additional insights if mRNA is still available. How these data match with previous Gohy's paper in the ERJ ?

'Referee Cross-Comments'

I agree with reviewer #2 that eventual differences in samples obtained from cancer vs lung transplant should be tested at least once statistically

Reviewer #2 (Comments to the Authors (Required)):

This is an interesting manuscript, that adds important information on top of other studies that demonstrated that features of the COPD airway epithelium in vivo persist upon in vitro culture. The inclusion of two groups of COPD patients that differ in severity, the inclusion of two control groups (non-smokers and smokers), as well as the systematic analysis of assessment of these features up to 10 weeks in ALI culture, offer important new insights. I have the following comments:

1. For obvious reasons, tissue was obtained through surgery for lung cancer, except for severe COPD where lung transplant explants were used. The relevance of these (inevitable) differences in characteristics of the subgroups should be discussed.

2. Somewhere in the discussion the authors should mention that both airway disease and emphysema are features in COPD patients, and that their relative contribution differs among patients.

3. Introduction, line 121. The reference to disease memory "(31)" should be adapted.

4. Introduction and Discussion. Please explain why these inflammatory stimuli for Figure 7 were selected, and why this may be more relevant than smoke exposures.

5. Results. In my opinion the authors use the word "questioned" when they actually mean "investigated" or "explored". Please check.

6. Results, line 158: the term "Epithelial pre-differentiation" is vague.

7. Have the authors evaluated whether the effect of exposure of the cultures to the cytokines on e.g. TEER persists upon removal of the cytokine?

8. Discussion. Some more attention should be paid to the relevance of EMT in the COPD lung in vivo.

9. Discussion. The following paper on basal cell heterogeneity in COPD may be relevant for the Discussion: Rao W, Wang S, Duleba M, Niroula S, Goller K, Xie J, Mahalingam R, Neupane R, Liew AA, Vincent M, Okuda K, O'Neal WK, Boucher RC, Dickey BF, Wechsler ME, Ibrahim O, Engelhardt JF, Mertens TCJ, Wang W, Jyothula SSK, Crum CP, Karmouty-Quintana H, Parekh KR, Metersky ML, McKeon FD, Xian W. Regenerative Metaplastic Clones in COPD Lung Drive Inflammation and Fibrosis. Cell 2020; 181(4): 848-864 e818

10. Also some discussion of epigenetic memory in the airway epithelium in COPD may be relevant.

11. Figure title: For clarity, I would suggest to not use abbreviations like "Smo" and "NS" in the figure legend title (fine in the graph legend and remainder of the legend).

Reviewer's comments – point by point response.

Reviewer #1 (Comments to the Authors (Required)):

We thank Reviewer 1 for her/his constructive comments. Point-by-point responses are provided herebelow.

Many thanks for giving me the opportunity to assess this study. I am impressed by the amount of experiments and data. Finally, it seems that airway epithelial features of COPD in ALI can be more and more accurately described. There are a few issues to address

1/ I am quite convinced that the culture medium have the potential to totally change the phenotype of the airway epithelium in ALI. Accordingly, I consider mandatory explaining in details how cultures were conducted.

Of course, this is an important point, as it is now clear that the culture medium may change the cellular composition of the reconstituted epithelium at the air/liquid interface (see for instance (Leung, Wadsworth et al., 2020)). Exhaustive details are already provided in the Expanded View Content. To make it clearer, we have added the following sentence in the main manuscript: "Extensive information on how epithelial cultures at the ALI were conducted is available in the Expanded View Content" (lines 362-363 of the revised manuscript).

However, although culture medium may modify the airway epithelial phenotype in ALI, all cultures have been conducted in the same manner, and it is therefore unlikely that differences observed across the study groups (i.e., COPD versus controls) may result from the choice of the culture medium.

2/ Correlations with FEV1 are shown for TEER, occludin, foxJ1, secretory component, ... This is a bit confusing actually, given the different fields explored. This is moreover confusing that these correlations are always in the positive way. The message associated is not necessarily supporting the message defended by the authors - that I share, and I would suggest rethinking about the hypothesis behind that. Elsewhere, FEV1 actually is potentially not the marker we'd like to see as a very reliable surrogate marker in COPD.

We thank Reviewer 1 for this interesting comment. Indeed, most correlations observed in our study are positive. In other words, most readouts assessed in our study are less expressed with disease severity. In our opinion, this is mainly due to the fact that COPD is a "dedifferentiating" disease at the epithelial level, where all epithelial properties (barrier function, polarity, ciliated cells differentiation) seem to be reduced in severe disease. Nonetheless, we looked for negative correlations (for fibronectin and vimentin, for instance), but these were not significant or non-existent.

Secondly, we acknowledge that the FEV1 is probably not the best marker to describe airway obstruction. The FEV1/FVC ratio or airway resistance parameters (sRaw) are probably more relevant. However, the current clinical definition of COPD severity still refers to FEV1 (cf. GOLD guidelines 2023), and we chose to follow this clinical definition to make our translational research more accessible to clinicians.

If requested, we should be able to explore correlations with FVC/FEV1, while sRaw measurements are not available for all patients included.

3/ EMT is more and more clearly seen as a critical step for repair. I am not totally sure that fibronectin is really reflecting that. Maybe PCR of ZEB/TWIST can provide additional insights if mRNA is still available. How these data match with previous Gohy's paper in the ERJ ?

We thank Reviewer 1 for pointing this inaccuracy. Indeed, fibronectin has been reported to both induce EMT and to be increased by EMT-related transcription factors such as TWIST1, SNAIL1, SNAIL2 and ZEB. Therefore, it is not clear whether it is part of abnormal repair (EMT type II) in COPD.

Slight modifications have been brought to the manuscript to avoid misleading readers on this particular point, in results sections 'EMT of the COPD AE' (line 185) and 'Exogenous inflammation induces COPD-like AE features' (lines 226 of the revised manuscript) as well as in the discussion (lines 291 of the revised manuscript).

To enhance the precision of our conclusions, we investigated the mRNA levels of ZEB and TWIST1, as suggested by Reviewer 1, alongside SNAIL1 and SNAIL2, which are other transcription factors implicated in EMT. Our findings revealed an increase in all four transcription factors among COPD patients during early, short-term, and occasionally mid-term ALI cultures. However, this heightened expression appeared to diminish in long-term ALI cultures. These results, now incorporated into the manuscript (lines 191-196 and 289-296 of the revised manuscript) and represented in a dedicated panel (Figure 5D), substantiate our overarching conclusions. Specifically, they affirm that EMT exhibits heightened activity in COPD (and smoker) samples for up to 4 weeks in ALI cultures but diminishes over time.

Globally, these data confirm the observations of Gohy and colleagues in their paper 2015 in the ERJ. Indeed, they observed increased fibronectin release in COPD ALI cultures as compared to controls, that was fading away over time. Furthermore, the dispersion of fibronectin measurements was quite wide, as is the case in our study.

'Referee Cross-Comments'

I agree with reviewer #2 that eventual differences in samples obtained from cancer vs lung transplant should be tested at least once statistically.

See response in Reviewer 2 point-by-point responses.

Reviewer #2 (Comments to the Authors (Required)):

This is an interesting manuscript, that adds important information on top of other studies that demonstrated that features of the COPD airway epithelium in vivo persist upon in vitro culture. The inclusion of two groups of COPD patients that differ in severity, the inclusion of two control groups (non-smokers and smokers), as well as the systematic analysis of assessment of these features up to 10 weeks in ALI culture, offer important new insights.

We thank Reviewer 2 for her/his comments and suggestions. Point-by-point responses are provided herebelow.

I have the following comments:

1. For obvious reasons, tissue was obtained through surgery for lung cancer, except for severe COPD where lung transplant explants were used. The relevance of these (inevitable) differences in characteristics of the subgroups should be discussed.

Indeed, this is an important point to discuss. We have added the following sentences in the limitations paragraph (lines 332-340 of the revised manuscript): “Third, initial samples were either collected from patients undergoing lobectomy (NS, Smo and COPD1-2 groups, in most cases for oncologic indication) or from explants issued from patients undergoing lung transplantation (and therefore free from cancer, COPD3-4 group). Although the samples were collected as far as possible from the tumor, an effect of distant cancer on the results cannot be ruled out. However, the results most likely to be positively influenced by a cancer (EMT, barrier dysfunction) are the most altered in the COPD3-4 group, free of cancer. On the other hand, most of the observed alterations in the COPD3-4 group are already present to a lesser extent in the COPD1-2 group, making an effect related to group disparity highly unlikely.”

As suggested by Reviewer 1, we also added statistical analysis in Table 1 and S1.

2. Somewhere in the discussion the authors should mention that both airway disease and emphysema are features in COPD patients, and that their relative contribution differs among patients.

The following sentence has been added in the first introduction paragraph : “These changes lead to airway obstruction (bronchitis) and emphysema, with their relative contribution varying among individuals diagnosed with COPD.” (lines 76-78 of the revised manuscript).

3. Introduction, line 121. The reference to disease memory "(31)" should be adapted.

We thank Reviewer 2 for pointing this out. The reference has been updated. (now line 122 of the revised manuscript).

4. Introduction and Discussion. Please explain why these inflammatory stimuli for Figure 7 were selected, and why this may be more relevant than smoke exposures.

First, we aimed at exploring whether epithelial inflammation could promote COPD-like features. Therefore, IL-6, IL-1b and TNF-alpha were chosen because these cytokines were clearly reported as being produced by the airway epithelium (see (Kany, Vollrath et al., 2019)) and increased in the BAL and/or serum of COPD patients. In this regard, IL-8/CXCL-8 could have been chosen too, but its production was mostly reported by neutrophils, that are absent in our epithelial cultures at the ALI.

Second, inflammatory cytokines were chosen rather than smoke exposure because we aimed at exploring the effect of COPD-related inflammation rather than acute cigarette smoke-related inflammation, which may vary from one another.

More pragmatically, the cigarette smoke exposure system was not available in our facility. Cigarette-smoke extract could have been added, but it does not accurately reproduce luminal exposure.

5. Results. In my opinion the authors use the word "questioned" when they actually mean "investigated" or "explored". Please check.

The word “questioned” was replaced as suggested (2 occurrences in the manuscript, lines 144 and 155 of the revised manuscript).

6. Results, line 158: the term "Epithelial pre-differentiation" is vague.

The term has been replaced by “Basal cells early differentiation” (line 158 of the revised manuscript)

7. Have the authors evaluated whether the effect of exposure of the cultures to the cytokines on e.g. TEER persists upon removal of the cytokine?

No, this was not explored due to the initial experimental design, which aimed at exploring the direct effect of inflammation on the epithelium. However, this suggestion is highly relevant, and attention

should be given to this point in future experiments. One could also suggest multiple short-term inflammatory stimulation phases mimicking exacerbation to match the clinical course of the disease and see how this influences the epithelial barrier. We feel this additional experiments are beyond the scope of the present manuscript.

8. Discussion. Some more attention should be paid to the relevance of EMT in the COPD lung in vivo.

The following sentences have been added to the discussion to illustrate the relevance of EMT in COPD: "EMT is a physiological process notably involved in tissue repair (type II EMT), which is dysregulated in COPD where it participates to airway fibrosis (Bartis, Mise et al., 2014). In addition, EMT is thought to play a role in the increased susceptibility to lung cancer in COPD individuals; however, further confirmation is required (Mahmood, Shukla et al., 2021)." (lines 289-292 of the revised manuscript).

9. Discussion. The following paper on basal cell heterogeneity in COPD may be relevant for the Discussion: Rao W, Wang S, Duleba M, Niroula S, Goller K, Xie J, Mahalingam R, Neupane R, Liew AA, Vincent M, Okuda K, O'Neal WK, Boucher RC, Dickey BF, Wechsler ME, Ibrahim O, Engelhardt JF, Mertens TCJ, Wang W, Jyothula SSK, Crum CP, Karmouty-Quintana H, Parekh KR, Metersky ML, McKeon FD, Xian W. Regenerative Metaplastic Clones in COPD Lung Drive Inflammation and Fibrosis. Cell 2020; 181(4): 848-864 e818

We appreciate Reviewer 2 for highlighting this important study. However, Rao and colleagues specifically isolated small airways progenitors, while our study focuses on large airways. The airway epithelium's cellular composition varies significantly along the bronchial tree, making it risky to extrapolate these findings from distal to proximal areas. Therefore, similar studies should be led on proximal airways progenitor cells, which is one of the goals of our research team in the next years.

10. Also some discussion of epigenetic memory in the airway epithelium in COPD may be relevant.

We thank Reviewer 2 for this highly relevant comment. nevertheless, the scarcity of data in this field prevents us to engage in such discussion.

Intriguingly, COPD basal cells, which we believe harbor the memory of epithelial damage in COPD, have not yet undergone specific assessment for epigenetic modifications in comparison to (non)-smokers. On one hand, cigarette smoke-induced epigenetic alterations, namely aberrant DNA methylation and histone modifications, have been widely described (Gao, Jia et al., 2015, Schamberger, Mise et al., 2014). On the other hand, the largest epigenetic study in COPD was performed on peripheral white blood cells, identifying the involvement of many pathways including inflammatory, wound healing, and stress response pathways (Qiu, Baccarelli et al., 2012). At the respiratory level, methylomic analyses have demonstrated COPD-related aberrant DNA methylation of genes involved in oxidative response pathways (Vucic, Chari et al., 2014), cell differentiation (Song, Heijink et al., 2017), and histone deacetylase function (Lam, Cloonan et al., 2013) but these results were obtained by analysing small airways brushings, primary epithelial cultures homogenates, and peripheral lung parenchyma homogenates, respectively, thereby preventing dedicated cell type analyses ("bulk epigenetics").

11. Figure title: For clarity, I would suggest to not use abbreviations like "Smo" and "NS" in the figure legend title (fine in the graph legend and remainder of the legend).

Corrections have been made as requested to ensure clarity for the readers.

References :

- Gao X, Jia M, Zhang Y, Breitling LP, Brenner H (2015) DNA methylation changes of whole blood cells in response to active smoking exposure in adults: a systematic review of DNA methylation studies. *Clin Epigenetics* 7: 113
- Kany S, Vollrath JT, Relja B (2019) Cytokines in Inflammatory Disease. *Int J Mol Sci* 20
- Lam HC, Cloonan SM, Bhashyam AR, Haspel JA, Singh A, Sathirapongsasuti JF, Cervo M, Yao H, Chung AL, Mizumura K, An CH, Shan B, Franks JM, Haley KJ, Owen CA, Tesfaigzi Y, Washko GR, Quackenbush J, Silverman EK, Rahman I et al. (2013) Histone deacetylase 6-mediated selective autophagy regulates COPD-associated cilia dysfunction. *J Clin Invest* 123: 5212-30
- Leung C, Wadsworth SJ, Yang SJ, Dorscheid DR (2020) Structural and functional variations in human bronchial epithelial cells cultured in air-liquid interface using different growth media. *Am J Physiol Lung Cell Mol Physiol* 318: L1063-I1073
- Qiu W, Baccarelli A, Carey VJ, Boutaoui N, Bacherman H, Klanderman B, Rennard S, Agusti A, Anderson W, Lomas DA, DeMeo DL (2012) Variable DNA methylation is associated with chronic obstructive pulmonary disease and lung function. *Am J Respir Crit Care Med* 185: 373-81
- Schamberger AC, Mise N, Meiners S, Eickelberg O (2014) Epigenetic mechanisms in COPD: implications for pathogenesis and drug discovery. *Expert Opin Drug Discov* 9: 609-28
- Song J, Heijink IH, Kistemaker LEM, Reinders-Luinge M, Kooistra W, Noordhoek JA, Gosens R, Brandsma CA, Timens W, Hiemstra PS, Rots MG, Hylkema MN (2017) Aberrant DNA methylation and expression of SPDEF and FOXA2 in airway epithelium of patients with COPD. *Clin Epigenetics* 9: 42
- Vucic EA, Chari R, Thu KL, Wilson IM, Cotton AM, Kennett JY, Zhang M, Lonergan KM, Steiling K, Brown CJ, McWilliams A, Ohtani K, Lenburg ME, Sin DD, Spira A, Macaulay CE, Lam S, Lam WL (2014) DNA methylation is globally disrupted and associated with expression changes in chronic obstructive pulmonary disease small airways. *Am J Respir Cell Mol Biol* 50: 912-22

December 7, 2023

RE: Life Science Alliance Manuscript #LSA-2023-02341-TR

Dr. Francois Michel Carlier
Université Catholique de Louvain
Institut de Recherche expérimentale et clinique
Avenue Mounier, 54
Claude Bernard Tower, 3rd floor
Brussels 1200
Belgium

Dear Dr. Carlier,

Thank you for submitting your revised manuscript entitled "The memory of airway epithelium damage in smokers and COPD patients". We would be happy to publish your paper in Life Science Alliance pending final revisions necessary to meet our formatting guidelines.

- please address Reviewer 2's remaining minor comment
- please upload your main and supplementary figures as single files
- please add the Twitter handle of your host institute/organization as well as your own or/and one of the authors in our system
- please use the [10 author names, et al.] format in your references (i.e. limit the author names to the first 10)
- please add a callout for Figure 5D to your main manuscript text
- please upload your Tables in editable .doc or excel format; -Tables should be numbered consecutively with Arabic numerals (1, 2, 3, 4); They can be included at the bottom of the main manuscript file or be sent as separate files.
- please incorporate the Supplemental Material into the main Material and Methods section. Any mention to the Supplemental Methods should be removed from the text. The supplemental Reference should also be incorporated into the main References list. We have no limits on these sections.

A. FINAL FILES:

B. MANUSCRIPT ORGANIZATION AND FORMATTING:

Sincerely,

Reviewer #1 (Comments to the Authors (Required)):

All my comments are addressed
I support publication of this manuscript

Reviewer #2 (Comments to the Authors (Required)):

Overall, I am pleased by the revision prepared by the authors based on the comments, and their thoughtful reply to the comments. I have an editorial comment on the text newly added to the Discussion:
Line 295: "where it participates to airway fibrosis" could be replaced by "where it is thought to contribute to airway fibrosis"

December 11, 2023

RE: Life Science Alliance Manuscript #LSA-2023-02341-TRR

Dr. Francois Michel Carlier
Université Catholique de Louvain
Institut de Recherche expérimentale et clinique
Avenue Mounier, 54
Claude Bernard Tower, 3rd floor
Brussels 1200
Belgium

Dear Dr. Carlier,

Thank you for submitting your Research Article entitled "The memory of airway epithelium damage in smokers and COPD patients". It is a pleasure to let you know that your manuscript is now accepted for publication in Life Science Alliance. Congratulations on this interesting work.

DISTRIBUTION OF MATERIALS:

Again, congratulations on a very nice paper. I hope you found the review process to be constructive and are pleased with how the manuscript was handled editorially. We look forward to future exciting submissions from your lab.

Sincerely,
